# Seroprevalence of West Nile Fever and Associated Risk Factors in Livestock of Afar Region, Northeast Ethiopia

**DOI:** 10.3390/vetsci12020141

**Published:** 2025-02-08

**Authors:** Jemberu Alemu Megenas, Mengistu Legesse Dadi, Tesfu Kassa Mekonnen, James W. Larrick, Gezahegne Mamo Kassa

**Affiliations:** 1Department of Veterinary Microbiology, Immunology and Public Health, College of Veterinary Medicine and Agriculture, Addis Ababa University, P.O. Box 34 Bishoftu, Ethiopia; 2College of Agriculture and Natural Resources, Gambella University, P.O. Box 126 Gambella, Ethiopia; 3Aklilu Lemma Institute of Pathobiology, Addis Ababa University, P.O. Box 1176 Addis Ababa, Ethiopia; 4Panorama Research Institute, Sunnyvale, CA 94089, USA; jwlarrick@gmail.com

**Keywords:** livestock, West Nile virus, seroprevalence, Afar region, Ethiopia

## Abstract

Our study assessed the seroprevalence of West Nile virus (WNV) infection in domestic animals in the Amibara and Haruka districts of Ethiopia’s Afar pastoral region, testing 736 serum samples from camels, cattle, donkeys, goats, and sheep. The overall seroprevalence of WNV IgG antibodies was 50.7%, with donkeys showing the highest prevalence (76.1%), followed by camels (69%), cattle (52%), goats (34.7%), and sheep (25.7%). These findings revealed a significantly higher prevalence compared with earlier studies in Ethiopia and other pastoral regions worldwide. Geographical differences, favorable vector breeding conditions, and temperature were identified as key factors influencing transmission dynamics. Risk factors such as species, sex, age, and location were examined, with species emerging as the most significant predictor of seropositivity. Female animals showed slightly higher seroprevalence, and older animals exhibited lower rates. The study underscores the importance of domestic livestock as sentinels for WNV surveillance, emphasizing the implications for both animal and human health in the region. These findings provide critical insights into the transmission of WNV, species-specific variations, and the environmental factors driving its prevalence in northeast Ethiopia.

## 1. Introduction

The Horn of Africa has historically been prone to mosquito-borne human and animal diseases [1]. Among others, mosquito-borne arboviruses such as West Nile virus pose an increasing global health concern as they are capable of rapidly spreading in new areas with suitable vectors and vulnerable hosts. According to Smithburn [2], West Nile virus (WNV) was originally isolated in Uganda’s West Nile district in 1937. Today, the virus is widely distributed across numerous countries, including Ethiopia. West Nile Virus (WNV) can infect multiple species, with birds acting as primary reservoirs. Humans and horses are also vulnerable, although they are dead-end hosts, meaning they do not contribute to further spread. This broad host range makes WNV a concern for both public health and animal care. The infection can manifest as asymptomatic, mild febrile illness, meningitis, or encephalitis or can even lead to fatalities, as highlighted by MacIntyre, Lourens [3]. As noted by Mohammed, Yasmin [4], WNV poses a significant risk to both human and animal health.

WNV is primarily maintained in nature through a cycle involving birds and mosquitoes. Multiple species of birds and mosquitoes support the replication of the virus. Clinical disease is primarily observed in horses and humans among mammals [5]. It is also maintained in enzootic cycles involving mosquitoes, mainly of the Culex mosquitoes, and birds as the reservoir hosts [6]. Humans and domesticated animals can also be affected by this virus, as mentioned by Alzuheir, Fayyad [7].

The virus spreads naturally between birds and mosquitoes, occasionally spilling over when infected Culex mosquitoes feed on humans, horses, ruminants, wildlife, and reptiles [8]. WNV infection has been extensively documented in horses, with some findings indicating it also exists in goats, sheep, and cattle [4].

Clinical illness resulting in neurological complications has been reported in these species on rare occasions [9]. Because of the virus’s low and transitory viremic phase during infection, these species act as dead-end hosts, neither multiplying nor transferring it at all. However, WNV infection in these species, particularly in horses, might result in severe losses for animal owners and the possibility of outbreaks affecting other species, including humans [9].

Due to Ethiopia’s tropical climate fostering a diverse range of flora and fauna, including mosquitoes such as Culex and various wild bird species, the Afar pastoral region experiences heightened activity and population of West Nile fever vectors [10]. Recent Ethiopian studies have indicated the presence of WNV in cattle, with seropositive rates of 5.4% in Gambella and 4.8% in South Omo, Ethiopia [11,12]. Despite an increase in information on WNV activity, there are currently no available data on WNV infection in equines and domesticated livestock populations. Consequently, this study aimed to assess exposure to WNV in cattle, sheep, goats, camels, and equines through serological methods.

## 2. Materials and Methods

### 2.1. Description of the Study Area

A cross-sectional study was undertaken in Ethiopia’s Afar region between June 2021 and April 2022 (Figure 1). The region is located in the northeastern half of the country and shares borders with Eritrea, Djibouti, and various Ethiopian districts. The Afar region, covering 72,052.78 square kilometers, had an estimated population of 1,945,801 in 2012, with a significant majority living in rural areas [13]. The region experiences high temperatures year-round, with an average annual range of 23 to 33 °C and peak temperatures reaching up to 45 °C. The primary livelihood of the inhabitants, who are predominantly engaged in pastoralism, relies on raising livestock, including camels and cattle, with natural pasturage as the main feed source. The study focused on two adjacent districts, Amibara and Haruka, in the southern part of the Afar region. These districts were chosen for their high cattle numbers, accessibility, and relative peace and security.

In June 2021, a preliminary community-based survey and observational study were conducted in the *Amibara* and *Haruka* districts to collect data on factors influencing West Nile fever virus (WNFV) infection. These districts were selected based on their livestock density, proximity to water bodies, extensive irrigation activities, and a history of flooding, all of which create favorable conditions for Arbovirus vector breeding. The study also examined instances of abortion and retained placenta in livestock within these districts.

### 2.2. Study Design

A cross-sectional study was carried out to evaluate the seroepidemiological status of WNV infection. The research utilized a convergent parallel mixed method, combining quantitative and qualitative data to conduct a thorough analysis of the research problem. This approach involved collecting both qualitative and quantitative types of data concurrently, followed by the integration of information during the interpretation of the overall results.

### 2.3. Study Animals

The investigation encompassed cattle, camels, goats, sheep, and donkeys within the two intentionally chosen districts of the Afar region. All settlements (villages) within each chosen sub-district were systematically incorporated into the study after securing consent from the elders, clan leaders, or kebele administrators to participate in the research based on the livestock population information.

### 2.4. Sample Size Determination and Sampling Method

A simple random sampling technique was utilized to select the study animals. The required sample size was determined using the sample size calculation formula outlined by Thrusfield (2005), as follows:$$n=\frac{1.96^{2}\times \mathrm{pexp}\left(1-\mathrm{pexp}\right)}{d^{2}}$$
where n = required sample size, pexp = expected prevalence = 50%, and d = desired absolute precision = 0.05.

In the absence of prior data on disease prevalence in the region, an expected prevalence of 50% was assumed, resulting in a calculated sample size of 384 per study district, with a total of 736 samples collected. Information such as age, sex, parity, herd size, history of mass abortion, mass death of young animals, and specific clinical signs for each animal was recorded individually using a provisional paper-based identification system. This information were gathered during blood sample collection using a checklist formatted as a brief questionnaire.

### 2.5. Sampling Procedure and Data Collection

With informed consent obtained from local elders and animal owners, animals within the herds were selected using a haphazard sampling method due to the lack of livestock registration in the study area. In the chosen households, interviews were conducted with individuals aged over 18 who had provided their informed consent.

A 5 mL blood sample was collected from chosen animals via jugular vein puncture using a sterile plain vacutainer tube. The obtained blood samples were labeled and stored at room temperature until the clot formed. Blood samples were centrifuged at 3000 rpm for 10 min to separate serum, which was subsequently kept at −20 °C.

### 2.6. Serological Analysis

Anti-WNFV-IgG antibodies were identified with the ID Screen^®^ WNV competition multispecies ELISA kits (ID-Vet Innovative Diagnostics, Montpellier, France). These kits have a specificity of 98.81% to 100% and a sensitivity of 83.89% to 100% for horse serum samples.

The results were measured at an optical density (OD) of 450 nm using a 96-well ELISA plate reader (Multiskan™ FC Microplate Photometer) and classified as positive or negative according to the cutoff values specified by the manufacturer. The test’s validity was confirmed in accordance with the manufacturer’s manual, which specified that validation occurred when the mean value of the negative control optical density (ODnc) exceeded 0.7 (ODnc > 0.7) and when the mean value of the positive control OD (ODPC) was less than 30% of the ODnc (ODPC/ODnc < 0.3). Then, the inhibition rate was calculated according to the following formula:$$\frac{S}{N}\left(\%\right)=\frac{ODs}{ODnc}\times 100$$
where OD is the optical density, nc is the negative control, S is the sample, and S/N is the competition percentage. S/N values lower than or equal to 40% were considered positive, values above 50% were considered negative, and values greater than 40% and less than or equal to 50% are finally considered negative.

### 2.7. Ethical Considerations

The Animal Research Ethics Committee at Addis Ababa University’s College of Veterinary Medicine and Agriculture thoroughly examined the study. Ethical approval (VM/ERC/40/03/15/2023) was obtained for all methods used, including the handling and collection of blood samples. Informed consent was secured from animal owners, and the study was conducted in compliance with the ARRIVE guidelines [14]. Ethical standards were rigorously followed, with a strong commitment to upholding the five principles of animal welfare. This comprehensive approach ensured the ethical integrity of the research and underscored the commitment to responsible and humane treatment of animals involved in the study.

### 2.8. Data Analysis

Data entry, cleaning, and validation were performed using Microsoft Office™ Excel^®^ 2019. The dependent variable in this study was the WNV ELISA test results (positive or negative). Inferential analyses were carried out using R software version 4.0.3 for Windows through R-Studio version 1.3.1093, alongside Microsoft Office™ Excel^®^ 2019. Descriptive and inferential analyses were also conducted using these tools. Mean differences between variables were analyzed using one-way ANOVA and Student’s *t*-test. Pearson’s Chi-square test of association was used in bivariate analysis, with Fisher’s exact test. The significance level was set at *p* < 0.05.

To analyze multicollinearity in WNV ELISA test results, univariable logistic regression models were built. Significant variables (*p* < 0.05) from the univariable regression analysis were included in the multivariable logistic regression model. The Hosmer–Lemeshow test was used to evaluate the model’s fit, while the omnibus test was used to examine its predictive performance.

## 3. Results

### 3.1. Description of Study Participants and Animals

A total of 87 owners of livestock participated in the study and provided written consent for the administration of a questionnaire and the collection of whole blood samples from their animals. Of the 87 participants, 68.97% participants were males, and 91.95% earned their living through subsistence farming, whereas 8.05% (7/80) engaged in other sources of revenue. A total of 736 (cattle, camel, goat, sheep, and donkey) blood samples were collected from the study animals (Table 1).

### 3.2. Seroprevalence of WNFV Infection

The seroprevalence of WNV infection was determined from 736 serum samples that were obtained from five different species of livestock populations (camel, cattle, donkey, goat, and sheep) that were located in two districts (Amibara and Haruka) of the Afar pastoral region. Of 736 tested livestock serum samples, 50.7% (373/736) exhibited IgG antibodies by competitive ELISA (95% CI: 47–54.4%; *p* < 0.05). The seroprevalence was higher (*p* < 0.05) in donkeys (76.1%) followed by camels (69.3%), cattle (52%), goats (34.7%), and sheep (25.7%). There is no statistical difference in the seroprevalence of WNV infection by age category or sex of tested animals (Table 2).

There was a statistical difference in the seroprevalence of WNV among species and districts of the study (Table 2).

Donkeys were (OR: 6.447, 95% CI = 3.888–10.688) seven times more likely to be seropositive for WNV infection than sheep (*p* < 0.01) (Table 3).

Respondents to the focus group discussions stated that “many of the available veterinary clinics and animal health posts were not providing the required functions due to the absence of the necessary utilities, livestock medical supplies and well-trained animal health workers”. Trained animal health professionals and para-veterinarians should be distributed throughout the areas to provide technical assistance to pastoralists and farmers. “The study districts have given due attention to expanding irrigation schemes as it has huge potential for irrigated agriculture which has favored proliferation of vectors of diseases such as West Nile fever and other zoonotic vector borne diseases”, according to a primary source of information.

## 4. Discussion

West Nile virus infection is traditionally considered “endemic” in Africa, particularly in sub-Saharan Africa, but the precise situation of the disease in Ethiopia has not been well established until now, and the obtained results allowed us to highlight the presence of IgG West Nile in livestock sera in the districts of Amibara and Haruka, a remote pastoral area. Indeed, only a few studies [11,12] have considered the virus’s existence in the Gambella and South Omo cattle populations, respectively.

In this study, a significant difference in seroprevalence was observed between species of the animals tested. The highest prevalence (76.1%) was noted in donkeys followed by camel (69.3%), cattle (52.2%), goat (34.7%), and sheep (25.7%). The current overall seroprevalence of WNV was significantly higher than that of earlier research conducted in Ethiopia, particularly in areas where pastoralism is the predominant mode of livelihood [11,12], and can have significant implications for both animal [15] and human health [16]. The reason for this relatively high seroprevalence among animals in the study area could be due to the conducive nature of mosquito breeding sites as samples were taken in close proximity to the Awash River basin, which causes frequent flooding, and its geographical location on migratory bird routes, which play an important role in the epidemiology of the virus.

The seroprevalence result proved that domestic livestock were implicated as useful sentinels for WNV, although the biological basis for this remains unknown [17]. These findings may be attributed to the domestic animals being present in large herds, which attract a greater number of mosquitoes, which is supported by Ulloa, Ferguson [18]. It is worth noting that WNV antibodies were found in the majority of domestic animals in the research districts, which may contribute to the virus’s maintenance and spread among animals.

From the present study, a higher 70/92 (76.1%) seroprevalence of IgG to WNV infection was documented in the donkey population, which was significantly higher than studies from Nigeria [19], Namibia [20], and Egypt [21] in Africa; Palestine and Israel [22]; and Turkey and Spain [23]. But the current prevalence of WNV infection was slightly less than findings from Sudan (88.75%) and Senegal (86.2%) [24].

In this study, 69% seropositivity of WNF IgG was detected in camels, which is considered to be high compared with other studies like in Nigeria (17.7%) [25], Egypt (40%) [17], Palestine (40%) [22], and Turkey (44%) [26].

Previous studies showed 4.8% [12] and 5.5% [11] seroprevalences of IgG to WNV infection in cattle from the South Omo and Gambella regions, respectively. In comparison with the above findings, the present study revealed significantly higher (52%) seroprevalence of WNV IgG, which might be due to geographical differences and vector activity. Similar study showed 22% seroprevalence in Egypt [17], 32.53% in Malaysia [4], and 20% in Turkey [26]. The discrepancy between the WNV seroprevalence rates reported in this study and the previous studies may also be partially attributable to differing sampling techniques.

A slightly higher (48.27%) seroprevalence of West Nile fever in goats was documented in Malaysia [4]. But the present study revealed that the seroprevalence of IgG to WNV in goats was higher (34.7%) than the seroprevalence in goats studied in Egypt (5.3%) [17] and Senegal (6.9%) [24].

The current work investigated the serological prevalence of antibodies against WNV in sheep; based on the cELISA test, it was 25.7% relatively lower than other domestic animals, but this result was higher than studies from Senegal (0%) [24], Egypt (3.5%) [17], Nigeria (20%) [27], and Turkey (0%) [26].

The spread and transmission of WNV are influenced by several factors. Weather is one of the drivers that affects vector competence, vector population dynamics, and virus replication rate in mosquitoes both directly and indirectly. Temperature has a significant impact on WNV transmission and viral replication rates [28].

In this investigation, the determining factors that were shown to be substantially linked with WNV exposure in the multivariable analysis were species differences in the study region, which was consistent with the findings of Paz [29,30].

Factors such as sex, age, species, and district were considered as potential risk factors for the seroprevalence of West Nile fever virus (WNV). A higher seroprevalence (51.2%) was observed in female animals, while a lower seroprevalence (50%) was found in older animals. The distribution of WNV seropositivity in this study was significantly associated with species (*p* < 0.01). Logistic regression analysis further confirmed that species was a significant risk factor for WNV seropositivity in the two districts.

## 5. Conclusions

The present study reports a notably high seroprevalence of IgG to WNV in the livestock population, with 50.7% of the sampled livestock testing positive. Unfortunately, there are no comparative national WNV data in Ethiopia due to the lack of an active WNV surveillance program. The possibility of arboviral epidemics is still a major worry as environmental and socioeconomic conditions change. To mitigate this risk, it is crucial for governments, healthcare organizations, and researchers to work collaboratively on surveillance, prevention, and response strategies. Consistent surveillance of WNV infection with prompt management of identified WNV disease in humans is of utmost importance.

## Figures and Tables

**Figure 1 vetsci-12-00141-f001:**
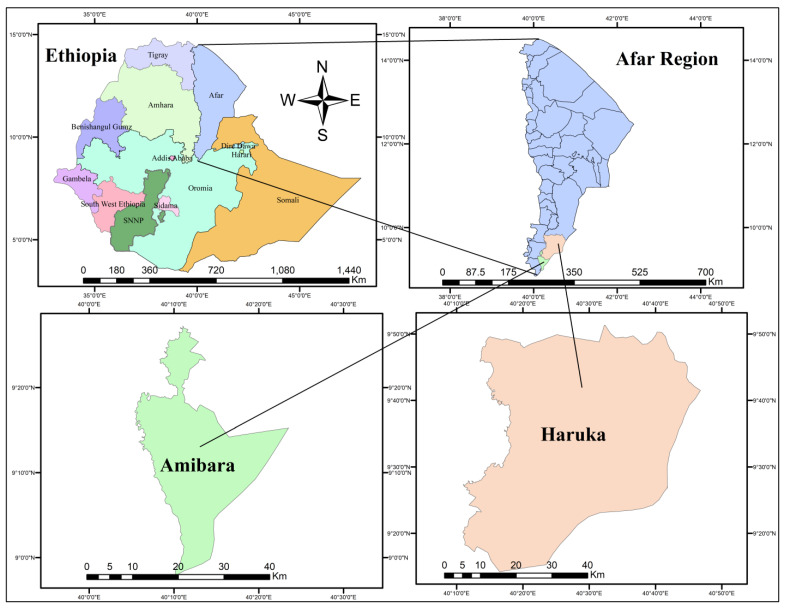
Map of the study area.

**Table 1 vetsci-12-00141-t001:** Profiles of the study animals.

Study Animals	Frequency	Proportion (%)
**Species**		
Camel	155	21.1
Cattle	224	30.4
Donkey	92	12.5
Goat	121	16.4
Sheep	144	19.6
**Sex**		
Female	582	79.1
Male	154	20.9
**District**		
Amibara	421	57.2
Haruka	315	42.8

**Table 2 vetsci-12-00141-t002:** The seroprevalence of WNFV in livestock populations across different variables.

Variable	Category	No. of Animals Tested	No. Positive	Prevalence (%)	95% CI	χ2	*p* Value
	Camel	155	107	69.3	60.2–75.4		
	Cattle	224	117	52.2	45.7–59.2		
Species	Donkey	92	70	76.1	65.7–84.2	93.171	0.000
	Goat	121	42	34.7	23.5–40.8		
	Sheep	144	37	25.7	20.2–35.3		
Sex	Female	582	298	51.2	46.6–57.0	0.305	0.322
	Male	154	75	48.7	39.5–55.3		
District	Amibara	421	216	51.3	46.7–57.2	0.155	0.694
	Haruka	315	157	49.8	37.5–53.5		
	<2 years	74	41	55.4	51.3–60.3		
Age	2 to <5 years	208	114	54.8	50.9–57.1	3.629	0.092
	5 to <10 years	332	157	47.3	44.2–52.4		
	>10 years	122	61	50.0	47.8–55.1		
Overall		736	373	50.7	47–54.4		

**Table 3 vetsci-12-00141-t003:** Univariate and multivariate regression analysis of risk factors associated with WNV infection seropositivity.

Variable	Category	No. of Sera Tested	No. of Positive Sera	Prevalence (%)	COR (95% CI)	AOR (95% CI)
Species	Camel	155	107	69.3	4.193 (2.528–6.955)	3.162 (1.973–5.067)
	Cattle	224	117	52.2	2.039 (1.327–3.133)	1.515 (1.038–2.212)
	Donkey	92	70	76.1	6.447 (3.888–10.688)	4.937 (3.076–7.925)
	Goat	121	42	34.7	0.701 (0.389–1.261)	0.477 (0.275–0.828)
	Sheep	144	37	25.7	Ref.	Ref.
Sex	Female	582	298	51.2	1.105 (0.775–1.577)	1.174 (0.802–1.718)
	Male	154	75	48.7	Ref.	Ref.
Age	<2 years	74	41	55.4	1.024 (0.601–1.747)	0.475 (0.318–0.707)
	2 to <5 years	208	114	54.8	1.385 (0.835–2.298)	0.576 (0.401–0.828)
	5 to <10 years	332	157	47.3	1.242 (0.656–2.218)	0.504 (0.314–0.811)
	>10 years	122	61	50.0	Ref.	Ref.
District	Amibara	421	216	51.3	1.006 (0.807–1.255)	1.086 (0.793–1.486)
	Haruka	315	157	49.8	Ref.	Ref.

## Data Availability

The data from this study have not been deposited into any public repository and will be made available upon request.

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
