# Peer review of "Seroprevalence of West Nile Fever and Associated Risk Factors in Livestock of Afar Region, Northeast Ethiopia"

_vetsci, 2025, doi:10.3390/vetsci12020141_

Round 1

Reviewer 1 Report

Comments and Suggestions for Authors

The following work describes the seroprevalence of WNV in the livestock population in Ethiopia. The work presents sufficient data for a publication; it is written quite well, although some aspects related to the epidemiological cycle of WNV should be explored in more depth. The data analysis is not convincing, in particular the calculation of the chi-square and the subsequent use of the OR (they should be calculated via logistic regression of only the variables significant in the chi-square). These methodological and structural weaknesses require major revisions. Below are my specific comments:
Abstract:
1) Line 11: Instead of "susceptible", I would say that the virus is "widespread".
2) Line 14: Explain why: in the face of symptoms and almost zero impact, they can play an important sentinel role to understand the evolution of epidemiology in other susceptible species such as horses and humans.
3) Line 16: What do the authors mean by "haphazard sampling"?
4) Check the spaces in the abstract and throughout the manuscript.
5) Why are only some risk factors mentioned in the abstract and not all those evaluated?
Introduction:
1) Line 53: It is necessary to differentiate the symptomatology and the role of equines from that of ruminants.
2) Other reports have also been carried out on wild animals such as wild boars and domestic animals such as dogs and cats (for example, in Campania, Italy; Canada, etc.). I believe it can be useful, and I advise the authors to cite these studies.

Materials and methods:
1) Some subsections can be included together.
2) Discuss (both here and in discussions) the fact that it is not possible to distinguish anti-West Nile antibodies and those of other related arboviruses and that probably the rates of WNV are increased by this (e.g., Usutu, from the assay datasheet, etc.).
3) Line 115: Was the calculation done for each animal species? What is the rationale for the subdivision between species?
Results:
1) Results of table 3 (but also of other tables) not mentioned in the abstract
2) I believe there are errors in the calculation of the chi-square; I ask the authors to double-check them (how is it possible that 51.3 vs. 49.8 is significant?).
3) The results of table 4 should be explained in the discussion. What hypothesis do the authors have for this similar trend in different species? Other studies?
4) Limit the multivariate analysis to significant variables.
Discussion:
1) The discussion is limited to buying the prevalences obtained in other African states. Having an international readability, I believe that studies carried out in other districts should also be compared (even over time).

Author Response

Comments and Suggestions for Authors

The following work describes the seroprevalence of WNV in the livestock population in Ethiopia. The work presents sufficient data for a publication; it is written quite well, although some aspects related to the epidemiological cycle of WNV should be explored in more depth. The data analysis is not convincing, in particular the calculation of the chi-square and the subsequent use of the OR (they should be calculated via logistic regression of only the variables significant in the chi-square). These methodological and structural weaknesses require major revisions. Below are my specific comments:
Abstract:
1) Line 11: Instead of "susceptible", I would say that the virus is "widespread." accepted 
2) Line 14: Explain why: in the face of symptoms and almost zero impact, they can play an important sentinel role to understand the evolution of epidemiology in other susceptible species such as horses and humans. We couldn't include horses and human samples in the study.
3) Line 16: What do the authors mean by "haphazard sampling"? Since there is no livestock registration in the study area, we took systematic sampling in the absence of livestock identification numbers. 
4) Check the spaces in the abstract and throughout the manuscript. accepted 
5) Why are only some risk factors mentioned in the abstract and not all those evaluated? only significant risk factors are mentioned. 
Introduction:
1) Line 53: It is necessary to differentiate the symptomatology and the role of equines from that of ruminants. okay 
2) Other reports have also been carried out on wild animals such as wild boars and domestic animals such as dogs and cats (for example, in Campania, Italy; Canada, etc.). I believe it can be useful, and I advise the authors to cite these studies. accepted 

Materials and methods:
1) Some subsections can be included together. accepted 
2) Discuss (both here and in discussions) the fact that it is not possible to distinguish anti-West Nile antibodies and those of other related arboviruses and that probably the rates of WNV are increased by this (e.g., Usutu, from the assay datasheet, etc.).
3) Line 115: Was the calculation done for each animal species? What is the rationale for the subdivision between species? Yes, the calculation is for each animal to see the differences in susceptibility.
Results:
1) Results of table 3 (but also of other tables) not mentioned in the abstract... if it is necessary, we will include it in the abstract.
2) I believe there are errors in the calculation of the chi-square; I ask the authors to double-check them (how is it possible that 51.3 vs. 49.8 is significant?). the statstics shows it is significant 
3) The results of table 4 should be explained in the discussion. What hypothesis do the authors have for this similar trend in different species? Other studies? okay..
4) Limit the multivariate analysis to significant variables. okay 
Discussion:
1) The discussion is limited to buying the prevalences obtained in other African states. Having an international readability, I believe that studies carried out in other districts should also be compared (even over time). 

Reviewer 2 Report

Comments and Suggestions for Authors

The manuscript "Seroprevalence of West Nile Fever and Associated Risk Factors in Livestock of Afar Region, Northeast Ethiopia" is well structured and interesting and addresses a very important issue for public health and livestock production.

However, it is not suitable for publication in its current form and some changes are needed.

Introduction: some sentences are repetitive, the bibliography could be expanded (e.g.: Line 51: Petruccelli A, Zottola T, Ferrara G, Iovane V, Di Russo C, Pagnini U, Montagnaro S. West Nile Virus and Related Flavivirus in European Wild Boar (Sus scrofa), Latium Region, Italy: a retrospective study. Animals (Basel). 2020 Mar 16;10(3):494. doi: 10.3390/ani10030494.

Castro-Scholten S, Caballero-Gómez J, Bravo-Barriga D, Llorente F, Cano-Terriza D, Jiménez-Clavero MÁ, Jiménez-Martín D, Camacho-Sillero L, García-Bocanegra I. Exposure to West Nile Virus in Wild Lagomorphs in Spanish Mediterranean Ecosystems. Zoonoses Public Health. 2024 Dec 18. doi: 10.1111/zph.13200.    Aguilera-Sepúlveda P, Cano-Gómez C, Villalba R, Borges V, Agüero M, Bravo-Barriga D, Frontera E, Jiménez-Clavero MÁ, Fernández-Pinero J. The key role of Spain in the traffic of West Nile virus lineage 1 strains between Europe and Africa. Infect Dis (Lond). 2024 Sep;56(9):743-758. doi: 10.1080/23744235.2024.2348633.   Adjadj NR, Vervaeke M, Sohier C, Cargnel M, De Regge N. Tick-Borne Encephalitis Virus Prevalence in Sheep, Wild Boar and Ticks in Belgium. Viruses. 2022 Oct 26;14(11):2362. doi: 10.3390/v14112362.   

Niczyporuk JS, Jabłoński A. Serologic Survey for West Nile Virus in Wild Boars (Sus scrofa) in Poland. J Wildl Dis. 2021 Jan 6;57(1):168-171. doi: 10.7589/JWD-D-19-00004.).

Results and discussion: Confidence intervals are recommended for all prevalence values. The tables are confusing, it is recommended to separate the variables (age, gender, etc.) by a line.

Comments on the Quality of English Language

English editing is recommended.

Author Response

Comments and Suggestions for Authors

The manuscript "Seroprevalence of West Nile Fever and Associated Risk Factors in Livestock of Afar Region, Northeast Ethiopia" is well structured and interesting and addresses a very important issue for public health and livestock production.

However, it is not suitable for publication in its current form and some changes are needed.

Introduction: some sentences are repetitive, the bibliography could be expanded (e.g.: Line 51: Petruccelli A, Zottola T, Ferrara G, Iovane V, Di Russo C, Pagnini U, Montagnaro S. West Nile Virus and Related Flavivirus in European Wild Boar (Sus scrofa), Latium Region, Italy: a retrospective study. Animals (Basel). 2020 Mar 16;10(3):494. doi: 10.3390/ani10030494. it has been corrected 

Castro-Scholten S, Caballero-Gómez J, Bravo-Barriga D, Llorente F, Cano-Terriza D, Jiménez-Clavero MÁ, Jiménez-Martín D, Camacho-Sillero L, García-Bocanegra I. Exposure to West Nile Virus in Wild Lagomorphs in Spanish Mediterranean Ecosystems. Zoonoses Public Health. 2024 Dec 18. doi: 10.1111/zph.13200.    Aguilera-Sepúlveda P, Cano-Gómez C, Villalba R, Borges V, Agüero M, Bravo-Barriga D, Frontera E, Jiménez-Clavero MÁ, Fernández-Pinero J. The key role of Spain in the traffic of West Nile virus lineage 1 strains between Europe and Africa. Infect Dis (Lond). 2024 Sep;56(9):743-758. doi: 10.1080/23744235.2024.2348633.   Adjadj NR, Vervaeke M, Sohier C, Cargnel M, De Regge N. Tick-Borne Encephalitis Virus Prevalence in Sheep, Wild Boar and Ticks in Belgium. Viruses. 2022 Oct 26;14(11):2362. doi: 10.3390/v14112362.   

Niczyporuk JS, Jabłoński A. Serologic Survey for West Nile Virus in Wild Boars (Sus scrofa) in Poland. J Wildl Dis. 2021 Jan 6;57(1):168-171. doi: 10.7589/JWD-D-19-00004.).

Results and discussion: Confidence intervals are recommended for all prevalence values. The tables are confusing; it is recommended to separate the variables (age, gender, etc.) by a line.

we will correct and add confidence interval soon 

Comments on the Quality of English Language....we hope we can manage it well 

English editing is recommended.

Reviewer 3 Report

Comments and Suggestions for Authors

The Editor Veterinary Sciences

Thank you for the opportunity to review the manuscript: “ Seroprevalence of West Nile Fever and Associated Risk Factors 2 in Livestock of Afar Region, Northeast Ethiopia”. The paper has been carefully reviewed but significant concerns arose:

In the introduction, there is information that is not directly associated with the species (humans or animals), causing difficulty in understanding. It is important to clarify whether the clinical signs observed are in humans or animals. Additionally, there are no clinical cases described in other species besides humans and equids. It is crucial for the author to clarify, even in the introduction, that the presence of antibodies is related to the circulation of the agent, not necessarily with the clinical presentation of the disease. Is there any information on the disease data in humans of this region?

 I did not find the specific correlation of the presence of WNV with the occurrence of abortions, especially since numerous other diseases can cause this symptomatology. On the other hand, the occurrence of sudden death cases is an important correlation.

The test used shows good specificity and sensitivity for equids. What guarantees its use and good results for other species?

From an epidemiological perspective and study characteristics, the term occurrence is more appropriate than prevalence.

Figures 2, 3 and 4 are unnecessary.

Information such as history of mass abortion, mass death of young animals, and specific clinical signs for each animal were obtained from the herd but was not used in the discussion at any point. It would also be important to know which avian orders (local and migratory) could participate in the cycle in the region.

Author Response

Comments and Suggestions for Authors

The Editor Veterinary Sciences

Thank you for the opportunity to review the manuscript: “ Seroprevalence of West Nile Fever and Associated Risk Factors  in Livestock of Afar Region, Northeast Ethiopia”. The paper has been carefully reviewed but significant concerns arose:

In the introduction, there is information that is not directly associated with the species (humans or animals), causing difficulty in understanding. It is important to clarify whether the clinical signs observed are in humans or animals. Additionally, there are no clinical cases described in other species besides humans and equids. It is crucial for the author to clarify, even in the introduction, that the presence of antibodies is related to the circulation of the agent, not necessarily with the clinical presentation of the disease. Is there any information on the disease data in humans of this region? in previous study results showed the presence of unknown febrile infections in human patients and most probably arboviral infections. 

 I did not find the specific correlation of the presence of WNV with the occurrence of abortions, especially since numerous other diseases can cause this symptomatology. On the other hand, the occurrence of sudden death cases is an important correlation.

The test used shows good specificity and sensitivity for equids. What guarantees its use and good results for other species? previous results seen from different studies.

From an epidemiological perspective and study characteristics, the term occurrence is more appropriate than prevalence.

Figures 2, 3 and 4 are unnecessary. we will manage it.

Information such as history of mass abortion, mass death of young animals, and specific clinical signs for each animal were obtained from the herd but was not used in the discussion at any point. It would also be important to know which avian orders (local and migratory) could participate in the cycle in the region. thank you we will incorporate 

Round 2

Reviewer 3 Report

Comments and Suggestions for Authors

 Thank you for the opportunity to review revised version of the manuscript: “ Seroprevalence of West Nile Fever and Associated Risk Factors 2 in Livestock of Afar Region, Northeast Ethiopia”. The paper has been carefully reviewed but significant concerns arose:

 Some suggestions were not accepted, or even responded to, such as:

 “The specificity and sensitivity test for other species.” The author mentions that there are other studies that use this technique on other species, but does not cite these studies in their literature review.

I still think figures 2, 3 and 4are unnecessary and should be removed from the work. Their inclusion has not been justified.

This question was not answered.”It would also be important to know which avian orders (local and migratory) could participate in the cycle in the region.

Author Response

 “The specificity and sensitivity test for other species.” The author mentions that there are other studies that use this technique on other species, but does not cite these studies in their literature review... the information was indicated in the leaflet of the manufactorers. 

I still think figures 2, 3 and 4are unnecessary and should be removed from the work. Their inclusion has not been justified.....removed 

This question was not answered.”It would also be important to know which avian orders (local and migratory) could participate in the cycle in the region...most probably migratory birds.